The potential, analysis and prospect of ctDNA sequencing in hepatocellular carcinoma

Ding Yubo 1 2
Yao Jingwei 1 2
Wen Meiling 3
Liu Xiong 4
Huang Jialu 2
Zhang Minghui 2
Zhang Yu 2
Lv Yufan 1
Xie Zhuoyi 2
Zuo JianHong 632138414@qq.com 1 2 5
1 The Affiliated Nanhua Hospital of University of South China , Hengyang, Hunan , China
2 University of South China, Transformation Research Lab, Hunan Province Key Laboratory of Tumor Cellular and Molecular Pathology, Hengyang Medical School , Hengyang, Hunan , China
3 The First Affiliated Hospital, University of South China , Hengyang , China
4 Department of Otolaryngology-Head and Neck Surgery, Nanfang Hospital, Southern Medical University , Guangzhou , China
5 Clinical Laboratory, The Third Affiliated Hospital of University of South China , Hengyang, Hunan , China
Chen Hao
Electronic publication date: 2022 May 17
Publication date: 2022
Volume: 10
Electronic Location ID: e13473
Received 2021 Oct 21; Accepted 2022 Apr 29
Copyright: ©2022 Ding et al.
Copyright year: 2022
Copyright holder: Ding et al.
License: This is an open access article distributed under the terms of the Creative Commons Attribution License, which permits unrestricted use, distribution, reproduction and adaptation in any medium and for any purpose provided that it is properly attributed. For attribution, the original author(s), title, publication source (PeerJ) and either DOI or URL of the article must be cited.
License URL: https://creativecommons.org/licenses/by/4.0/

Keywords: Liver cancer, Next-generation sequencing, Gene mutation, Pathway analysis, Circulating tumor DNA

Funding: Key Research Program from the Science and Technology Department of Ningxia Hui Autonomous Region, China 2019BFH02012 Key Research Program of Hunan Health Committee 20201909 National Natural Science Foundation of China 82173008 Program of Hengyang science and Technology Bureau 2017-1 2020-67 This work is supported by the Key Research Program from the Science and Technology Department of Ningxia Hui Autonomous Region, China (2019BFH02012); the Key Research Program of Hunan Health Committee (20201909); the National Natural Science Foundation of China (82173008); and the Program of Hengyang science and Technology Bureau (2017-1, 2020-67). The funders had no role in study design, data collection and analysis, decision to publish, or preparation of the manuscript.

==============================
Background

The genome map of hepatocellular carcinoma (HCC) is complex. In order to explore whether circulating tumor cell DNA (ctDNA) can be used as the basis for sequencing and use ctDNA to find tumor related biomarkers, we analyzed the mutant genes of ctDNA in patients with liver cancer by sequencing.

Methods

We used next-generation targeted sequencing technology to identify mutations in patients with liver cancer. The ctDNA from 10 patients with hepatocellular carcinoma (including eight cases of primary hepatocellular carcinoma and two cases of secondary hepatocellular carcinoma) was sequenced. We used SAMtools to detect and screen single nucleotide polymorphisms (SNPs) and insertion deletion mutations (INDELs) and ANNOVAR to annotate the structure and function of the detected mutations. Screening of pathogenic and possible pathogenic genes was performed using American College of Medical Genetics and Genomics (ACMG) guidelines. GO analysis and KEGG analysis of pathogenic and possible pathogenic genes were performed using the DAVID database, and protein–protein interaction network analysis of pathogenic and possible pathogenic genes was performed using the STRING database. Then, the Kaplan–Meier plotter database, GEPIA database and HPA database were used to analyse the relationship between pathogenic and possible pathogenic genes and patients with liver cancer.

Results

Targeted capture and deep sequencing of 560 cancer-related genes in 10 liver cancer ctDNA samples revealed 8,950 single nucleotide variation (SNV) mutations and 70 INDELS. The most commonly mutated gene was PDE4DIP, followed by SYNE1, KMT2C, PKHD1 and FN1. We compared these results to the COSMIC database and determined that ctDNA could be used for sequencing. According to the ACMG guidelines, we identified 54 pathogenic and possible pathogenic mutations in 39 genes in exons and splice regions of 10 HCC patients and performed GO analysis, KEGG analysis, and PPI network analysis. Through further analysis, four genes significantly related to the prognosis of liver cancer were identified.

Conclusion

In this study, our findings indicate that ctDNA can be used for sequencing. Our results provide some molecular data for the mapping of genetic variation in Chinese patients with liver cancer, which enriches the understanding of HCC pathogenesis and provides new ideas for the diagnosis and prognosis of HCC patients.

Introduction

HCC is the fifth most commonly occurring cancer in the world, and its mortality rate is third among global cancer deaths, most of which occur in the 50- to 60-year-old population (Torre et al., 2015). The standardized mortality rate ranks third among all kinds of malignant tumours, next to gastric cancer and oesophageal cancer (Budny et al., 2017). According to the 2018 global cancer statistics report, 84,100 new liver cancer cases occurred worldwide and 781,000 new deaths, accounting for 4.7% and 8.2% of the cancer incidence rate and mortality, respectively (Bray et al., 2018). The pathological types of liver cancer include the hepatocyte type, bile duct type and mixed cell type, among which the hepatocyte type accounts for approximately 70% (Massarweh & El-Serag, 2017). In Japan and parts of China, the primary risk factors for hepatic carcinoma are chronic hepatitis C virus (HCV) or hepatitis B virus (HBV) (Kulik & El-Serag, 2019). The incidence of hepatocellular carcinoma is increasing, with a particularly high incidence in sub-Saharan Africa and the southeast (Mohammadian, Allah Bakeshei & Mohammadian-Hafshejani, 2020). Although the prognosis and treatment of hepatocellular carcinoma have been improving over the years, the morbidity of HCC is still increasing. In the past 20 years, progress has been made in elucidating the mechanism of cancer, early diagnosis of disease and improving local and systemic treatment of HCC (Page et al., 2014).

The diagnosis of HCC can be achieved using ultrasound and tumour markers to monitor high-risk groups. There was a significant positive correlation between alpha fetoprotein (AFP) and HCC. However, studies have shown that AFP lacks adequate sensitivity. AFP does not increase in many patients with liver cancer, it can even be normal in patients with advanced liver cancer (Bruix & Sherman , 2011). At present, the European Liver Association and the American Association for the Study of Liver Diseases no longer recommend the determination of AFP levels for the diagnosis of liver cancer, and there are some doubts regarding the diagnostic sensitivity of AFP (Kim et al., 2016). The presence or absence of cirrhosis with tumour features, including vascular invasion and portal vein thrombosis, tumour size and alpha-fetoprotein, are important prognostic indicators of HCC, which affects treatment decisions and outcomes (Al-Shamsi et al., 2017).

The occurrence of HCC is a progressive, dynamic and multigene-regulated pathological process. Therefore, understanding the molecular mechanism of its pathogenesis and screening according to the relevant factors in high-risk groups can prevent and help with treating HCC. Well known environmental risk factors that may lead to potential cirrhosis include hepatitis B virus, hepatitis C virus, exposure to toxins (such as aflatoxin) and alcohol consumption. Specific gene mutations have been isolated for each cause of HCC (Forner & Reig, 2018). In recent years, to improve the prognosis of liver cancer, many researchers have been committed to exploring tumour biomarkers. For example, Wei Lu suggested that expression levels of TCF21 in HCC were significantly decreased and negatively correlated with the invasive progression of the disease. TCF21 may be a biomarker for predicting the prognosis of HCC (Lu et al., 2019). Qiu, Pan & Li (2020) found that expression levels of lncRNA LOC285194 in tumour tissues was significantly lower than that in adjacent normal tissues, and it was often downregulated in HCC. LncRNA LOC285194 expression is closely related to the occurrence, development, invasion and metastasis of tumours and has antitumour effects, which can be used as a potential target for the development of new therapies for liver cancer. Although many biomarkers have been proposed, the prognostic evaluation of HCC is very challenging in the clinic. In addition, it is necessary to identify new and specific biomarkers for this malignant tumour.

Next-generation sequencing (NGS) technology is a new gene screening, prognosis and diagnosis technology and an effective and acceptable clinical gene detection method (Chen et al., 2021). Targeted sequencing is based on high-throughput sequencing, which can simultaneously detect multiple gene mutation types (including point mutation, insertion/deletion, copy number change and so on) of multiple cancer species. It is suitable for the detection of any tissue sample or liquid sample. The sequencing includes the whole exon region of 560 genes and the hot spot region of the TERT gene promoter mutation. All genes were derived from thousands of classic studies, the cancer genome census, the authoritative commercial cancer panels, and most of the driving genes in the databases closely related to clinical medication guidance. High frequency mutation genes and susceptibility genes are from three classical reviews (Vogelstein et al., 2013; Rahman, 2014; Kandoth et al., 2013) and are supplemented and sorted by literature collection and reading. This method is based on the Agilent SureSelect targeted sequence capture system, combined with repeatedly optimized probe design and powerful capture efficiency. The target gene has high coverage and strong specificity, which can realize the accurate detection of gene mutations and accurately screen cancer-related mutations.

Materials and Methods

Patients and DNA extraction

Our study was approved by the ethics committee of Affiliated Nanhua Hospital, University of South China (approval No. 201812). We received written informed consent from the participants, and the participants agreed to the publication of the research data obtained from these collected samples. A total of 10 subjects were included in this study. Nine patients with primary hepatocellular carcinoma, aged 33–68 years, were enrolled from January 2019 to December 2019 in our hospital. The diagnostic criteria of primary liver cancer (meeting any of the following three criteria) was as follows: (1), Two typical imaging manifestations of liver cancer (ultrasound, CT, MRI or selective hepatic arteriography), with a lesion > two cm. (2), A typical imaging manifestation, lesions > two cm, AFP > 400 ng/ml. (3), Positive liver biopsy. Among the patients with primary liver cancer, 1 case was diagnosed by pathological biopsy, and the other 8 cases were diagnosed by clinical diagnosis. Data from two patients with liver metastases (one with gallbladder cancer and liver metastasis, one with rectal cancer and liver metastasis) were also collected. The diagnostic criteria were implemented according to the guidelines for the diagnosis and treatment of gallbladder cancer (2015 Edition) and the Chinese code for the clinical diagnosis and treatment of colorectal cancer. Patients with primary liver cancer complicated by other tumours (such as gastric cancer, lung cancer, cervical cancer, ovarian cancer and prostate cancer), human immunodeficiency virus infection or autoimmune liver disease, alcoholic liver disease, nonalcoholic fatty liver disease, history of other chronic liver diseases, and samples with haemolysis during the test were excluded. According to the inclusion and exclusion criteria, nine cases in the primary HCC group and two cases in the liver metastasis group were collected. However, one patient in the primary liver cancer group whose ctDNA was not qualified after blood extraction was excluded, while the rest qualified for study inclusion, leaving eight cases in the primary HCC group and two cases in the liver metastasis group. Five millilitres of venous blood was collected by a special nurse using an EDTA anticoagulant tube. After balancing, it was placed into a centrifuge (within half an hour of blood drawing) and centrifuged at room temperature for 5 min. After centrifugation, plasma was collected and stored in a two ml EP tube in an ultralow refrigerator at −80 °C until ctDNA was extracted. A GeneRead DNA FFPE Kit (Qiagen) was used to extract ctDNA from plasma following the instructions of the kit. Agarose gel electrophoresis was used to analyse the extent of DNA degradation and whether there was contamination with RNA or protein, and Qubit was used to quantify the DNA concentration (DNA samples with DNA concentrations above 20 ng/µl and 0.2 µg above 0.2 µg were used to build the database).

Target sequencing

Nuohe Zhiyuan used Agilent’s liquid chip capture system to efficiently enrich human specific target region DNA and then conducted high-throughput and high-depth sequencing on the Illumina HiSeq platform. The Agilent SureSelectXT Custom kit was used for the database building and capture experiments. Only the reagents and consumables recommended in the manual were used, and the latest optimized experimental process was referred to for operation. Genomic DNA fragments 180–280 bp in length were randomly interrupted by a Covaris fragmentation apparatus. After terminal repair and addition of an A-tail, the ends of the fragments were connected to prepare the DNA library. After the library with a specific index was pooled, it was hybridized with up to 500,000 biotin-labelled probes in the liquid phase, and then the target gene fragment was captured using magnetic beads with streptomycin. After PCR linear amplification, the quality of the library was tested, and the qualified library was sequenced. After building the library, we used Qubit 2.0 for preliminary quantification, diluted the library to 1 ng/µL, and then used an Agilent 2100 to determine the insert size of the library. After the insertion size reached the expected value, the Q-PCR method was used to accurately quantify the effective concentration of the library (the effective concentration of the library > two nm) to ensure the quality of the library. The Illumina-HiSeq platform was sequenced according to the library’s effective concentration and data output requirements. The metadata were deposited in the NCBI Sequence Read Archive under accession no. PRJNA772098.

Data analysis

The raw image data files obtained by sequencing were filtered by raw reads for low quality, base uncertainty and other factors to obtain clean reads. The wrong sequencing data were screened out by the sequencing Phred value of each base. The quality of the sequencing data was primarily above Q20. The effective sequencing data were aligned to the reference genome (B37) using BWA and Samblaster, and the initial alignment results in Bam format were obtained. The BAM file was marked and repeated by Samblaster to obtain the final comparison result of the BAM format. ANNOVAR software was used to annotate the mutation sites, including gene structure annotation, genome feature annotation, nonsynonymous mutation hazard prediction, known mutation database annotation and mutation-related gene function annotation. RefSeq and Gencode were used to annotate the gene structure of the mutation site, including mRNA, noncoding RNA, small RNA and microRNA. The genomic characteristics of the mutation sites included CG Island, cell karyotype, phastconselements46way conserved region, genome repeat, transcription factor binding site and encode annotation of Gm12878 cell lines. SIFT, Polyphen, Mutationassessor, LRT and other methods were used to comprehensively evaluate the impact of nonsynonymous mutations on disease/tumour. dbSNP, thousand human genome SNP database, HapMap database, cosmic known tumour somatic mutation database and esp6500 mutation database were provided to screen any combination of mutation results. GO biological process, GO cell components, GO analysis function, KEGG, Reactome, Biocarta, PID and other functional annotation databases were used to interpret signal transduction and metabolic pathways. The Kaplan–Meier plotter database (http://www.kmplot.COM/analysis/) was used to analyse the prognosis with respect to the selected genes, and the survival curve was drawn using the Kaplan–Meier method. GEPIA (http://gepia.CancerPKU.CN/) was used to analyse the difference in gene expression between HCC and normal tissues. The HPA database (https://www.proteinatlas.org/) was used to verify the protein expression of genes in HCC and normal tissues.

Results

Clinical characteristics of patients with liver cancer

Ten patients with liver cancer were identified, including eight primary liver cancer patients, one liver metastasis of gallbladder carcinoma and one liver metastasis of rectal cancer. Table 1 summarizes the clinical and pathological data of all patients in this study. The clinical indicators include average age, sex, age range, stage, tumour size, lymph node, distant metastasis, AFP, hepatitis, and cirrhosis. As shown in Table 1, in eight cases of primary liver cancer, six cases had hepatitis, five of which had HBV. HBV is closely related to liver cancer, which was also confirmed by subsequent results.

Table 1 Clinicopathologic characteristics of patients with liver cancer.

Type		Primary liver cancer	Metastatic liver cancer	
Variable		No of patients N = 8 (%)	No of patients N = 2 (%)	
Name of cancer species		Primary hepatocellular carcinoma	Gallbladder carcinoma complicated with liver metastasis	Rectal cancer complicated with liver metastasis	
Age (years)	Mean ± SD	54 ± 10.32	/	/	
Range	33–68	55	64	
Gender	Male	6	1	1	
Female	2	/	/	
Clinical stage	I	0%	/	/	
II	0%	/	1(100)	
III	2(25)	/	/	
	IV	6(75)	1(100)	/	
Tumor size	T1	0%	/	/	
T2	1(12.5)	1(100)	1(100)	
T3	7(87.5)	/	/	
Lymph nodes status	N0	3(37.5)	/	1(100)	
N1	5(62.5)	1(100) /	
N2	0%	/	/	
Distant metastasis	M0	4(50)	/	1(100)	
M1	4(50)	1(100)	/	
Hepatitis	Positive	6(75)	/	/	
Negative	2(25)	1(100)	1(100)	
Liver cirrhosis	Positive	5(62.5)	/	/	
Negative	3(37.5)	1(100)	1(100)	
AFP	Positive	6(75)	1(100)	1(100)	
Negative	2(25)	/	/	
Notes.

AFP, alpha fetal protein

Gene mutation spectrum

We sequenced the whole exon region of 560 genes in 10 patients with liver cancer and the promoter mutation hotspot of the TERT gene. These 560 genes are hotspot genes in cancer research and clinical treatment related genes, including 194 clearly functional drivers, 127 cancer signalling pathway genes (12 genes enriched by KEGG pathway There are 86 high frequency mutation genes, 113 genes, 245 drug sensitivity genes, drug resistance related genes and 15 drug metabolism related genes. Four authoritative drug-related databases (Drugbank, MyCancerGenome, PharmGKB and PG_FDA) and manually sorted drug resistance genes were used for annotation, of which 202 genes were labelled drug target genes by at least one drug database, and 43 genes were labelled drug resistance genes. Fifteen genes related to drug metabolism were identified by drug metabolism pathway enrichment.

The target capture depth sequencing of 560 tumour-related genes in 10 ctDNA samples of hepatoma showed that 8940 SNV mutations were found in 481 genes, while only 19 genes had 67 insertions and deletions. There were 281 gene mutations over 10 and 145 gene mutations over 20. Figures S1 and S2 describe the number of SNPs and indels in different regions of each human genome (left) and the number of different types of SNPs and indels in coding regions (right). Table S1 describes all mutated genes in the targeted sequencing of HCC patients. Among them, the most commonly mutated genes were PDE4DIP (450), followed by SYNE1 (214), KMT2C (119), PKHD1 (110), FN1 (105), LRP1B (98), ALK (97), FANCA (95), NOTCH1 (92), ABCC4 (79), RNF213 (79), HNF1A (77), and LAMA2 (76). We analysed the SNV changes detected and found that in all patients with liver cancer, the changes of G > A, T > C, A > G, and C > T were more common than other changes, as shown in Fig. 1A.

Figure 1 Targeted cell SNVs by next-generation sequencing.

(A) Percentages of different transitions and transitions in SNV. (B) Proportion of SNV types by gene region. SNVs, single nucleotide variants.

In addition, we analysed the regions of these SNVs. Among these variants (Fig. 1B), exon variants (80.91%) were most common, followed by intron variants (13.31%), splicing variants (2.51%), and UTR variants (1.21%). In addition, we assessed the genetic effects of these variations in the exon region, including missense variation, synonymous mutation, stop gain/loss mutation and unknown mutation.

Analysis of circulating tumour DNA high-frequency gene mutations in patients with primary liver cancer

We detected a large number of cancer-related gene mutation sites and then compared these genes to the hepatoma top 20 mutant gene in the COSMIC database. Among the high-frequency genes of liver cancer, no mutation sites were found in six genes (PREX2, ZFHX3, FHIT, CAMTAL, GPHN and SND1), and the other 14 genes in the samples are shown in Table 2. According to the detected mutations, most of the high-frequency mutant genes of HCC in the cosmic database have mutation sites in the ctDNA of patients with primary liver cancer, but the mutation frequency is not exactly the same as the reference frequency in the COSMIC database, which may be related to the insufficient amounts of experimental samples and may be affected by other factors (somatic release) in the blood fluid of the body.

Table 2 Mutations of HCC high-frequency mutations in patients with primary liver cancer.

Sample	TP53	CTNNB1	AXIN1	LRP1B	ARID2	KMT2C	ALK	ERBB4	PTPRT	ZNF521	PTPRK	NTRK3	FOXP1	LPP	
1	+	-	+	+	+	+	+	+	+	-	+	+	+	-	
2	+	-	+	+	+	+	+	+	+	-	+	+	-	+	
3	+	-	-	+	+	+	+	+	+	-	+	+	-	+	
4	-	-	+	+	-	+	+	-	+	+	+	+	-	+	
5	-	+	+	+	+	+	+	+	+	-	+	+	+	+	
6	+	-	-	+	-	+	+	-	+	+	+	+	-	+	
7	+	+	+	+	-	+	+	-	+	-	+	+	+	+	
8	+	-	+	+	-	+	+	+	+	-	-	+	-	+	
Mutation frequency	75%	21%	75%	100%	50%	100%	100%	62.5%	100%	25%	90%	100%	37.5%	12.5%	
Reference frequency	28%	19%	9%	28%	12%	18%	17%	19%	16%	18%	17%	17%	14%	15%	
Notes.

The blue mutant genes represent the significant genes after sample screening.

TP53 and FOXP1 are mutation sites screened by ACMG guidelines, indicating that these mutations may be related to the development of primary liver cancer. They have diagnostic significance, providing possible drug treatment targets, and have further research significance.

Analysis of the gene mutation rate of high-frequency mutation genes in circulating tumour DNA from patients with secondary liver cancer

In patients with secondary liver cancer, 12 high frequency mutation genes were detected in liver cancer tissues with mutation sites, as shown in Table 3. Compared to primary liver cancer, FOXP1 had no mutation sites in secondary liver cancer, indicating it has high diagnostic specificity in primary liver cancer. In the gene samples from patients with secondary liver cancer, KMT2C/TP53 had significance after screening by ACMG guidelines. The high-frequency mutation genes of gallbladder cancer and rectal cancer were sought in the COSMIC database, in which KMT2C was the high-frequency mutation gene of gallbladder cancer and TP53 was the high-frequency mutation gene of rectal cancer, which was in line with the DNA gene mutation of circulating tumours in the blood. Therefore, KMT2C may be used as a reference gene for the diagnosis of gallbladder cancer, and TP53 is a reference gene for the diagnosis of colorectal cancer. These results indicate that ctDNA can also be used for sequencing.

Table 3 HCC high-frequency mutation in patients with liver metastases.

Sample	TP53	CTNNB1	AXIN1	LRP1B	ARID2	KMT2C	ALK	ERBB4	PTPRT	ZNF521	PTPRK	NTRK3	FOXP1	LPP	
1	+	–	–	+	+	+	+	+	+	+	+	+	–	+	
2	+	–	+	+	–	+	+	+	+	+	+	+	–	+	
Frequency	28%	19%	9%	28%	12%	18%	17%	19%	16%	18%	17%	17%	14%	15%	
Notes.

The blue mutant genes represent the significant genes after sample screening.

Comparison of high-frequency mutations between primary and secondary liver cancer

We compared the high-frequency mutant genes between eight cases of primary liver cancer and two cases of secondary liver cancer and found that there was no significant difference between the two groups (Table S2).

Pathogenic genes and possible pathogenic genes in filtrate

Based on NGS data, we further screened and analysed genes by Sorting Intolerant From Tolerant (SIFT), PolyPhen, Mutation Taster and Combination Annotation Dependent Deletion (CADD), zygosity, variation type, variation effect, location and filter coverage rate, small allele frequency and other conditions according to the guidelines of the American College of Medical Genetics and Genomics (ACMG). We found 39 mutated genes may be related to the development of liver cancer. The most frequently mutated genes were RET and TP53 (40%; 4/10), LAMA2 (30%; 3/10), and KAT6B, DNMT1, FGFR3, PRKDC, FOXP1, PDGFRB and SYNE1 (20%; 2/10).

We also observed that some mutation sites did not exist in the dbSNP database or COSMIC database. Ninety significant nonsynonymous mutation were identified, including 72 missense mutations, two deletion mutations, 1 code shifting deletion mutation, eight nonshift code deletion mutations, and seven nonshift code insertion mutations, which were located in 87 genes. After we summarized these loci, we extracted representative genes, which were AXIN1, CHD7, FN1, LAMA2, NIN, PRKDC, CEPPA, PTCH1 (2/10), PDE4DIP (4/10), and SYNE1 (3/10)). The mutation of these genes has further significance for the development of cancer.

GO notes and KEGG analysis on the filtered pathogenic genes or possible pathogenic genes

We used DAVID to analyse the Gene Ontology (GO) and Kyoto Encyclopedia of Genes and Genomes (KEGG) pathways. The GO term is annotated in three categories: cell component (GO-CC), molecular function (GO-MF) and biological process (GO-BP). CC is generally used to describe the role of gene products. For example, a protein may be located in the nucleus or ribosome. MF primarily refers to the tasks performed by gene product molecules. For example, a protein may be a transcription factor or a carrier protein. BP refers to a large biological function associated with gene products, such as mitosis. The KEGG access database (https://www.kegg.jp/kegg/) was used to annotate the enriched pathways. The GO terms and KEGG pathway were considered statistically significant at P < 0.05. According to ACMG guidelines, we identified 54 pathogenic or possibly pathogenic mutations in 39 genes in exons and splicing regions in 10 patients with liver cancer. (Table 4 and Table S3). We performed gene ontology annotation or pathway analysis on 39 pathogenic genes or possible pathogenic genes (Fig. 2 and Table S4).

Table 4 Summary of Likely Pathogenic mutations based on ACMG guidelines.

CHROM	POS	ID	REF	ALT	QUAL	FILTER	GeneName	Func	
14	92505923	rs186074112	T	A	228	PASS	TRIP11	exonic	
17	7577497	.	A	T	221	PASS	TP53	splicing	
17	7578291	.	T	G	221	PASS	TP53	splicing	
17	7578275	.	G	A	222	PASS	TP53	exonic	
17	7577507	.	T	A	222	PASS	TP53	exonic	
9	120475742	.	C	G	27.2252	PASS	TLR4	exonic	
6	152469504	rs267600861	C	T	221	PASS	SYNE1	exonic	
6	152557969	.	G	T	221	PASS	SYNE1	exonic	
12	111886081	rs199803113	T	C	222	PASS	SH2B3	exonic	
10	43601830	rs34682185	G	A	196	PASS	RET	exonic	
10	43620335	rs17158558	C	T	222	PASS	RET	exonic	
10	43601830	rs34682185	G	A	221	PASS	RET	exonic	
10	43620335	rs17158558	C	T	221	PASS	RET	exonic	
8	145741602	rs34633809	C	T	222	PASS	RECQL4	exonic	
13	49037865	.	A	G	222	PASS	RB1	splicing	
8	48694956	rs55769154	A	G	222	PASS	PRKDC	exonic	
8	48802944	.	A	G	222	PASS	PRKDC	exonic	
19	50910318	rs201804732	C	T	222	PASS	POLD1	exonic	
5	149512332	rs200684708	G	A	221	PASS	PDGFRB	exonic	
5	149512332	rs200684708	G	A	222	PASS	PDGFRB	exonic	
1	164776885	.	G	T	222	PASS	PBX1	exonic	
9	134027192	rs190788992	A	G	222	PASS	NUP214	exonic	
9	139395201	.	A	C	24.0543	PASS	NOTCH1	exonic	
6	129419442	.	C	T	222	PASS	LAMA2	exonic	
6	129465086	.	A	G	221	PASS	LAMA2	exonic	
6	129837463	rs200796753	G	T	216	PASS	LAMA2	exonic	
10	76780433	.	G	A	222	PASS	KAT6B	exonic	
10	76789416	rs140992439	C	T	222	PASS	KAT6B	exonic	
19	17943465	.	G	A	21.7998	PASS	JAK3	exonic	
6	407575	.	C	T	222	PASS	IRF4	exonic	
2	209101885	rs566072712	C	T	222	PASS	IDH1	exonic	
3	71021303	rs76145927	T	C	222	PASS	FOXP1	exonic	
3	71021303	rs76145927	T	C	222	PASS	FOXP1	exonic	
5	180030278	rs201796032	C	T	222	PASS	FLT4	exonic	
4	1805473	rs188723332	G	A	222	PASS	FGFR3	exonic	
4	1804706	.	C	A	222	PASS	FGFR3	exonic	
3	10140458	.	G	C	151	PASS	FANCD2	exonic	
16	89846317	rs201323171	C	T	222	PASS	FANCA	exonic	
13	103510736	rs56255799	C	T	222	PASS	ERCC5,BIVM-ERCC5	exonic	
19	45868349	rs150000483	C	T	213	PASS	ERCC2	exonic	
12	56478854	.	G	A	222	PASS	ERBB3	exonic	
17	37868208	.	C	T	222	PASS	ERBB2	exonic	
9	439392	rs117109271	A	G	222	PASS	DOCK8	splicing	
19	10260337	rs74505694	G	A	222	PASS	DNMT1	splicing	
19	10257115	rs550380640	G	A	222	PASS	DNMT1	exonic	
16	50827496	rs199624138	A	G	222	PASS	CYLD	exonic	
16	68867343	rs142927667	G	A	222	PASS	CDH1	exonic	
11	119155731	rs373989524	C	T	222	PASS	CBL	exonic	
2	202150044	.	A	C	221	PASS	CASP8	splicing	
10	88678970	.	C	A	222	PASS	BMPR1A	exonic	
22	23653939	.	C	T	45.6877	PASS	BCR	exonic	
5	112175271	.	C	A	176	PASS	APC	exonic	
1	179086577	.	A	C	137	PASS	ABL2	exonic	
2	169851916	.	A	T	222	PASS	ABCB11	exonic	

The results of GO annotation showed that the BP of these overlapping genes may be related to positive regulation of transcription of RNA polymerase II promoter, phosphorylation of peptide tyrosine, positive regulation of transcription, positive regulation of GTPase activity, signal transduction, MAPK cascade, negative regulation of apoptosis process, protein phosphorylation, and signal pathway of transmembrane receptor protein tyrosine activator. The results of MF annotation indicate that these genes are involved in protein binding, ATP binding, protein tyrosine kinase activity, transcription factor activity, sequence-specific DNA binding, RAS guanosine exchange factor activity, transcription factor binding, enzyme binding, protein kinase activity, receptor binding, and protein heterodimerization activity. In addition, the CC involved in these pathogenic or possibly pathogenic gene mutations was primarily related to the cytoplasmic perinuclear region, receptor complex, cell adhesion connection, cytoskeleton, extracellular components of the cytoplasmic side, replication fork, flotillin complex and catenin complex. These genes are involved in the occurrence and development of cancer in patients with liver cancer.

Further enrichment analysis based on the KEGG database showed that these pathogenic and possibly pathogenic genes were highly enriched in liver cancer and cancer-related pathways, as shown in Table 5 and Table S5. The approaches include cancer-related pathways, such as central carbon metabolism and protein polysaccharides, small RNAs in cancer, and various cancers, including thyroid, prostate and pancreatic cancers. These genes have also been found to be associated with HBV, which is one of the most related diseases to primary liver cancer.

Figure 2 GO enrichment analysis of 39 pathogenic and potential pathogenic genes.

(A) Results from GO analysis are presented in the bar plot (Top 10); (B) findings of the GO analysis are presented in the bubble chart (top 10).

Table 5 Pathway annotation of pathogenic genes and possible pathogenic genes.

Category	Term	Count	%	PValue	Genes	
KEGG_PATHWAY	hsa05200:Pathways in cancer	12	30	7.99E−07	BCR, RB1, PDGFRB, RET, CASP8, APC, LAMA2, CDH1, ERBB2, CBL, FGFR3, TP53	
KEGG_PATHWAY	hsa05219:Bladder cancer	5	12.5	3.46E−05	RB1, CDH1, ERBB2, FGFR3, TP53	
KEGG_PATHWAY	hsa05230:Central carbon metabolism in cancer	5	12.5	2.02E−04	PDGFRB, RET, ERBB2, FGFR3, TP53	
KEGG_PATHWAY	hsa05206:MicroRNAs in cancer	8	20	2.74E−04	PDGFRB, DNMT1, NOTCH1, ERBB3, APC, ERBB2, FGFR3, TP53	
KEGG_PATHWAY	hsa03420:Nucleotide excision repair	4	10	0.00129	POLD1, ERCC2, ERCC5, BIVM-ERCC5	
KEGG_PATHWAY	hsa05213:Endometrial cancer	4	10	0.001731	APC, CDH1, ERBB2, TP53	
KEGG_PATHWAY	hsa05218:Melanoma	4	10	0.004217	RB1, PDGFRB, CDH1, TP53	
KEGG_PATHWAY	hsa05220:Chronic myeloid leukemia	4	10	0.004387	BCR, RB1, CBL, TP53	
KEGG_PATHWAY	hsa04151:PI3K-Akt signaling pathway	7	17.5	0.004556	PDGFRB, LAMA2, FLT4, FGFR3, TP53, JAK3, TLR4	
KEGG_PATHWAY	hsa05166:HTLV-I infection	6	15	0.005867	RB1, PDGFRB, APC, POLD1, TP53, JAK3	
KEGG_PATHWAY	hsa04012:ErbB signaling pathway	4	10	0.007438	ERBB3, ERBB2, ABL2, CBL	
KEGG_PATHWAY	hsa05215:Prostate cancer	4	10	0.007676	RB1, PDGFRB, ERBB2, TP53	
KEGG_PATHWAY	hsa05216:Thyroid cancer	3	7.5	0.007871	RET, CDH1, TP53	
KEGG_PATHWAY	hsa05205:Proteoglycans in cancer	5	12.5	0.013183	ERBB3, ERBB2, CBL, TP53, TLR4	
KEGG_PATHWAY	hsa04550:Signaling pathways regulating pluripotency of stem cells	4	10	0.026606	APC, FGFR3, JAK3, BMPR1A	
KEGG_PATHWAY	hsa05223:Non-small cell lung cancer	3	7.5	0.02762	RB1, ERBB2, TP53	
KEGG_PATHWAY	hsa05161:Hepatitis B	4	10	0.029124	RB1, CASP8, TP53, TLR4	
KEGG_PATHWAY	hsa05212:Pancreatic cancer	3	7.5	0.036354	RB1, ERBB2, TP53	
KEGG_PATHWAY	hsa05214:Glioma	3	7.5	0.036354	RB1, PDGFRB, TP53	
KEGG_PATHWAY	hsa05222:Small cell lung cancer	3	7.5	0.058932	RB1, LAMA2, TP53	
KEGG_PATHWAY	hsa05203:Viral carcinogenesis	4	10	0.068735	RB1, CASP8, TP53, JAK3	
KEGG_PATHWAY	hsa04510:Focal adhesion	4	10	0.069536	PDGFRB, LAMA2, FLT4, ERBB2	
KEGG_PATHWAY	hsa04015:Rap1 signaling pathway	4	10	0.072781	PDGFRB, CDH1, FLT4, FGFR3	
KEGG_PATHWAY	hsa04014:Ras signaling pathway	4	10	0.086436	PDGFRB, FLT4, ABL2, FGFR3	
KEGG_PATHWAY	hsa05145:Toxoplasmosis	3	7.5	0.092188	CASP8, LAMA2, TLR4	

Pathway analysis and PPI protein prediction

The protein–protein interaction (PPI) network forecast was provided using STRING online software (https://string-db.org/). The PPI network of 39 pathogenic or possibly pathogenic genes was constructed using the STRING database, as shown in the Fig. 3. The string map shows that these genes contain 39 nodes and 126 edges. Network nodes represent proteins, and each node represents all the proteins produced by a single, protein-coding gene locus. Edges represent protein–protein associations, and proteins jointly contribute to a shared function. However, this does not necessarily mean that they physically bind to each other. It can be seen from the figure that proteins of other genes interact with each other except ABCB11 and LAMA2.

Figure 3 PPI networks of 39 pathogenic and potential pathogenic genes.

Survival analysis and expression of pathogenic and possible pathogenic genes

We used Kaplan–Meier plotter to analyse survival with respect to 39 pathogenic and possible pathogenic genes. The results showed that 22 of the 39 genes were correlated with OS (P < 0.05). The GEPIA database was used to analyse the expression levels of 22 genes in HCC and normal liver tissues. As shown in Fig. 4, expression levels of four genes (PRKDC, FANCD2, POLD1, and RECQL4) were significantly upregulated in HCC tissues (P < 0.05). Survival analysis (Fig. 5) showed that high expression of the PRKDC, FANCD2, POLD1, and RECQL4 genes was associated with reduced OS (P < 0.05). These results suggest that these four genes may play a role in the occurrence and development of HCC.

Figure 4 Expression of PRKDC, FANCD2, POLD1 and RECQL4 in HCC and normal tissues (GEPIA).

(A) Expression of FANCD2 in hepatocellular carcinoma and normal liver tissues. (B) Expression of POLD1 in hepatocellular carcinoma and normal liver tissues. (C) Expression of PRKDC in hepatocellular carcinoma and normal liver tissues. (D) Expression of RECQL4 in hepatocellular carcinoma and normal liver tissues.

Figure 5 Survival curves of four differential genes (Kaplan–Meier plotter).

(A) Survival curve of FANCD2 gene. (B) Survival curve of POLD1 gene. (C) Survival curve of PRKDC gene. (D) Survival curve of RECQL4 gene.

The HPA database was used to preliminarily verify the protein expression of the PRKDC, FANCD2, POLD1 and RECQL4 genes in HCC and normal tissues (Fig. 6). They were all highly expressed in HCC tissues compared to normal tissues, and PRKDC and RECQL4 have been proven to be related to poor prognosis in HCC (Qi et al., 2020; Li et al., 2018).

Figure 6 Immunohistochemistry (HPA): expression of PRKDC, FANCD2, POLD1 and RECQL4 in HCC and normal tissues.

(A) Expression of FANCD2 in hepatocellular carcinoma and normal liver tissues. (B) Expression of POLD1 in hepatocellular carcinoma and normal liver tissues. (C) Expression of PRKDC in hepatocellular carcinoma and normal liver tissues. (D) Expression of RECQL4 in hepatocellular carcinoma and normal liver tissues.

Discussion

In this study, a large number of liver cancer-related mutation genes were detected using ctDNA extraction, and the levels of ctDNA were significantly increased, which was similar to the contrast gene, in agreement with the experimental results. Most of the high frequency mutation genes in the HCC group exhibited mutation sites, which indicates that the mutation genes in blood ctDNA are basically the same as those in liver cancer tissues. It is of diagnostic value to verify that blood ctDNA does carry the information of the mutation gene in the liver cancer tissue.

In this study, 560 tumour-related genes from 10 hepatoma samples were sequenced in depth. We identified 8950 SNVs and 70 INDELs in 481 genes, among which PDE4DIP, SYNE1, KMT2C, PKHD1, FN1, LRP1B, ALK, FANCA, NOTCH1, ABCC4, RNF213, HNF1A, LAMA2, GF2R, NIN, PIK3C2B, APC and DOCK8 had more than 70 mutations. Thirty-nine pathogenic genes and possible pathogenic genes were identified by ACMG. Thirty-nine pathogenic genes and possible pathogenic genes were identified by ACMG aetiology analysis, and then these genes were analysed by GO enrichment analysis, pathway analysis and PPI network analysis. The Kaplan–Meier plotter database was used for survival analysis, the GEPIA database was used to analyse expression differences of genes between HCC and normal liver tissues, and the HPA database was used to analyse the expression of proteins in cancer and normal tissues to determine the relationship between these genes and clinical characteristics.

The mutation analysis was used to assess genetic variation of liver cancer, and it was found that PDE4DIP mutations were the highest in almost all patients with liver cancer, followed by SYNE1, KMT2C, PKHD1 and FN1. PDE4DIP is a protein-coding gene. The protein encoded by this gene anchors phosphodiesterase 4D to the dictyosome/centrosome region of the cell, participates in microtubule dynamics, promotes microtubule assembly, and plays a role at the level of the dictyosome or centrosome (Yang et al., 2017; Bouguenina et al., 2017). Gene defects may be one of the reasons for the correlation between myelodysplasia (MBD) and eosinophil proliferation and may also be one of the risk factors for adult pineal blastoma (Chang et al., 2017; Snuderl et al., 2018). The results of the database analysis showed that the PDE4DIP protein may interact with PDE4D, PRLAR2A and AKAP9. These proteins can be combined with cAMP or PKA, suggesting that PDE4DIP may participate in the cAMP/PKA signalling pathway. Our study suggests that PDE4DIP may be involved in the pathogenesis of liver cancer. In addition, SYNE1 is a coding gene that encodes an anchor protein that is expressed in skeletal muscle, smooth muscle and peripheral blood lymphocytes, is located in the nuclear membrane, and participates in the relationship between the nuclear layer and cytoskeleton (Indelicato et al., 2019). It has been reported that SYNE1 has higher transcriptional expression in the beginning and progression stages of HCC. The protein plays a potential role in the development of HCC and tumours (Faraj Shaglouf et al., 2020). KMT2C is a protein-coding gene that is a member of the ASC-2/NCOA 6 complex (AsCOM); it has histone methylation activity and participates in transcriptional CO activation (Cho et al., 2018). It has been reported that KMT2C is frequently mutated in a variety of human cancers and is crucial for the occurrence and development of most cancers (Chen et al., 2019). PRKDC is a member of the PI3/pi4 kinase family. PRKDC encodes the catalytic subunit of DNA-dependent protein kinase (DNA-PK) and plays a role in DNA double-strand break repair and recombination together with the Ku 70/Ku 80 heterodimer protein. Expression of PRKDC has been significantly correlated with the overall survival rate of HCC (Chaplin et al., 2021). The FN1 gene encodes fibronectin, a glycoprotein in the form of a soluble dimer in plasma and a dimer or polymer in the cell surface and extracellular matrix (Qi et al., 2020; Liu et al., 2020). Fibronectins are involved in cell adhesion, cell motility, opsonization, wound healing, and maintenance of cell shape (Cai et al., 2018). Fibronectin is a known biomarker for the inchoate diagnosis of HCC, and its changes may be an alternative indicator for evaluating the response of patients with early HCC after therapy (Kim et al., 2020). Significant mutations in SYNE1, KMT2C, PKHD1 and FN1 were observed in all 10 HCC patients.

We used the method based on the ACMG mutation classification guidelines in 481 mutated genes and identified 54 pathogenic and possible pathogenic mutations in 39 genes, which may be the cause of liver cancer in these individuals. GO and KEGG analyses were performed to further understand the role of pathogenic mutant genes. GO annotation revealed that BP included peptide tyrosine phosphorylation, positive regulation of RNA polymerase II promoter transcription, positive regulation of GTPase activity, protein autophosphorylation, phosphatidylinositol phosphorylation and so on, which may provide a basis for further understanding of the occurrence and development of liver cancer. In addition, the enriched KEGG pathway was found to be involved in Hepatitis B (hsa05161)- and tumour (hsa05200, hsa05230, hsa05206, hsa05205)-related pathways, and BCR, RB1, PDGFRB, RET, CASP8, APC, LAMA2, CDH1, ERBB2, CBL, FGFR3, and TP53 played a key role in carcinogenesis and tumour progression.

Through the analysis of Kaplan–Meier plotter, GEPIA and HPA databases, it was found that high expression of PRKDC, FANCD2, POLD1 and RECQL4 genes is significantly correlated with shorter OS, while FANCD2 and POLD1 have not been reported to be associated with liver cancer, which may become biological markers to evaluate the prognosis of HCC patients and provide new ideas for the diagnosis and treatment of HCC.

Conclusion

Inevitably, there are several shortcomings in this study. Above all, this is a straightforward centre research and requires a large-scale multiagency study to verify these results. Second, the potential functions and pathways were predicted using bioinformatics, which requires experimental verification. Third, because our sample size was small and ctDNA sequencing was used, there may be errors. However, our target sequencing can be used to efficiently, quickly and conveniently screen out hotspot genes of liver cancer and provide new ideas for the diagnosis and prognosis of patients with liver cancer.

In conclusion, we sequenced liver cancer samples, identified some new mutated gene sites and screened some genes that are significant for the diagnosis and prognosis of liver cancer. Our results provide some molecular data for the mapping of genetic variation in Chinese hepatoma patients and enrich the understanding of the pathogenesis of HCC. In addition, we screened some pathogenic genes according to ACMG guidelines, performed GO analysis, pathway analysis and PPI network analysis, and analysed their relationship with the clinic according to the Kaplan–Meier plotter, GEPIA and HPA databases to provide new ideas for the prognosis, treatment and diagnosis of liver cancer patients. Notwithstanding, further cohort research and experimental research are needed to explore the potential mechanism of HCC.

Supplemental Information

Supplemental Information 1 All SNP and INDEL mutated genes and their frequency

Click here for additional data file.

Supplemental Information 2 Comparison of high frequency mutation between primary and secondary liver cancer (TOP10)

Click here for additional data file.

Supplemental Information 3 Summary of Likely Pathogenic mutations based on ACMG guidelines

Click here for additional data file.

Supplemental Information 4 GO Annotation for pathogenic and likely pathogenic genes

Click here for additional data file.

Supplemental Information 5 Pathway annotation of pathogenic genes and possible pathogenic genes(KEGG)

Click here for additional data file.

Supplemental Information 6 Distribution of the number of SNPs in different regions of the genome (left) and the number of different types of SNPs in the coding region (right)

Click here for additional data file.

Supplemental Information 7 Distribution of INDEL number (left) in different regions of genome and different types of indel number (right) in coding region

Click here for additional data file.

Supplemental Information 8 Anonymised raw data for clinicopathologic characteristics

Click here for additional data file.

Supplemental Information 9 Mutation sequence

Click here for additional data file.

We thank all the authors for their contributions and support and the study participants who provided blood samples. We also thank the hospital staff for their contribution to the data collection for this study. We thank all programs and organizations for their support

Abbreviations

INDEL insertion deletion

SNP single nucleotide polymorphism

ctDNA circulating tumor cell DNA

HCC hepatocellular carcinoma

ACMG American College of Medical Genetics and Genomics

HCV hepatitis C virus

HBV hepatitis B virus

AFP alph fetoprotein

GO gene ontology

CC cell component

MF molecular function

BP biological process

KEGG Kyoto Encyclopedia of Genes and Genomes

SIFT Sorting Intolerant From Tolerant

CADD Combined Annotation Dependent Depletion

PPI proteinprotein interaction

Additional Information and Declarations

Competing Interests

Author Contributions

Human Ethics

DNA Deposition

Data Availability

The authors declare there are no competing interests.

Yubo Ding conceived and designed the experiments, performed the experiments, analyzed the data, prepared figures and/or tables, authored or reviewed drafts of the paper, and approved the final draft.

Jingwei Yao conceived and designed the experiments, performed the experiments, analyzed the data, prepared figures and/or tables, authored or reviewed drafts of the paper, and approved the final draft.

Meiling Wen analyzed the data, authored or reviewed drafts of the paper, and approved the final draft.

Xiong Liu performed the experiments, authored or reviewed drafts of the paper, and approved the final draft.

Jialu Huang analyzed the data, authored or reviewed drafts of the paper, and approved the final draft.

Minghui Zhang analyzed the data, authored or reviewed drafts of the paper, and approved the final draft.

Yu Zhang analyzed the data, authored or reviewed drafts of the paper, and approved the final draft.

Yufan Lv analyzed the data, authored or reviewed drafts of the paper, and approved the final draft.

Zhuoyi Xie analyzed the data, authored or reviewed drafts of the paper, and approved the final draft.

JianHong Zuo conceived and designed the experiments, authored or reviewed drafts of the paper, and approved the final draft.

The following information was supplied relating to ethical approvals (i.e., approving body and any reference numbers):

Our study was approved by the ethics committee of Affiliated Nanhua Hospital, University of South China (approval No. 201812).

The following information was supplied regarding the deposition of DNA sequences:

The sequences are available at NCBI Sequence Read Archive: PRJNA772098.

The following information was supplied regarding data availability:

The raw measurements are available in the Supplementary Files.

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
