# Peer review of "The potential, analysis and prospect of ctDNA sequencing in hepatocellular carcinoma"

_PeerJ, doi:10.7717/peerj.13473_

## Round 0.1 · original submission · Major Revisions

Dear Dr. Zuo,

My apologies for the long delay but we now have all three reviews. The second review is short but the remaining two reviews provided detailed comments that I hope will be useful for your revision.

Sincerely,

Hao Chen

·

Basic reporting

Please see my additional comments.

Experimental design

Please see my additional comments.

Validity of the findings

Please see my additional comments.

Additional comments

The manuscript entitled “The potential, analysis and prospect of ctDNA sequencing in hepatocellular carcinoma” by Yubo Ding et al. investigated that gene mutations using circulating tumor DNA (ctDNA) of 10 patients with hepatocellular carcinoma. They sequenced 560 cancer-related genes and compared their results with the cosmic database. This study is of potential clinical significance with some issues.
1. The two metastatic samples did not originate from hepatocellular carcinoma. They are metastatic tumors from gallbladder cancer and rectal cancer. Therefore, the statement of “2 cases of metastatic hepatocellular carcinoma” needs revising to avoid confusion.
2. The manuscript can be improved by moderate English and formatting editing. For example, “According to the global cancer statistics report in 2018, there are 841000 new liver cancer patients and 781000 deaths, accounting for 4.7% and 8.2% of the total cancer Morbidity and Mortality: in the world respectively”, “we found that 6 gene loci (PREX2, ZFHX3, FHIT, CAMTAL, GPHN and SND1) in the high frequency gene did not find mutation sites”, “FOXP1 did not find mutation site, only gene mutation occurred in primary liver cancer, and the specificity was relatively high in the diagnosis of liver cancer”, and “The pathway enrich of is applied to making pathway annotations”.
3. The resolution of Figure 1 needs improving.
4. The paragraph describing pathway analysis in text line 236 was misplaced. It should be under the subtitle “Go notes on the filtered pathogenic genes and possible pathogenic genes”.
5. What is the interpretation of GO annotations of pathogenic genes? How are they related to pathogenesis of hepatocellular carcinoma?
6. The description of Figure 3 were extremely succinct. What is the interpretation? What is the indication of “44 nodes and 143 edges”? Authors should elaborate and discuss.
7. The conclusion and significance of this study are overstated.
1) The study only sequenced 10 patients with hepatocellular carcinoma in one city of China, so it cannot provide “the gene mutation map of Chinese patients with liver cancer”.
2) Are the pathogenic mutations detected by ctDNA sequencing associated with patients’ treatment response, clinical stage, or prognosis? Are the mutations enriched in cancer samples compared to normal tissues? Without these data, authors cannot “confirms that ctDNA can be used for sequencing, and enriches the understanding of the pathogenesis of HCC” or “identified some new mutation gene sites and screened some genes which are of significance for the diagnosis and prognosis of liver cancer”.
3) With only one sample for each cancer, it cannot be concluded that “KMT2C/TP53 may be related to the development of gallbladder cancer / rectal cancer. KMT2C may be used as a reference gene for the diagnosis of gallbladder cancer, and TP53 is a reference gene for the diagnosis of colorectal cancer”.

Reviewer 2 ·

Basic reporting

The title and abstract appropriately reflect the main aspect of the work. The methods are clear and replicable. However, the samples size may be too small.

Experimental design

Experimental design meet the standards。

Validity of the findings

The final data is plausible but the results are not novel and unlikely to be of interest to readers.

Additional comments

There is need for checking grammar.

·

Basic reporting

The English language should be improved by a fluent speaker to ensure that an international audience can clearly understand your text. Some examples where the language could be improved include results (lines 196 to 206), conclusion part (lines 319 to 329) et al– the current phrasing makes comprehension difficult.
Figures are not well labelled and illustrated with poor resolution quality.

Experimental design

(1) A diagram was better than text to show the study design.
(2) Since HBV infection is closely related to the pathogenesis of primary hepatocellular carcinoma, is there any association on gene mutation spectrum with/without HBV infection?
(3) To confirm ctDNA could be used for sequencing, it is important to find proper controls such as tumor tissue samples. Or it would be clinical significance if there is correlation between gene spectrum with clinical-pathological parameters. Otherwise, it is not meaningful to just list out all the mutations by sequencing.
(4) The results part is long and not comprehensively. Since there are 8 patients with primary hepatocellular carcinoma and 2 patients with metastases liver cancer, it would be better to compare spectrum of Mutations of high-frequency mutations between those two groups.

Validity of the findings

Improvement in next-generation sequencing technology and better understanding of genetic or epigenetic alteration of primary hepatocellular carcinoma have allowed comprehensive analysis of mutational landscape of ctDNA. Hotspot mutation panels have both shown promising performance which have been validated in current study as well. But none of these tests including current study have yet been validated. More large-scale tests in longitudinal cohorts are needed for preclinical detection of liver cancers.

---

## Round 0.2 · Minor Revisions

Dear Dr. Zuo,

Please see comments by Reviewer 2 on improving the language throughout the manuscript, including the abstract, but also please improve the results and discussion sections to make the description more concise.

Hao Chen

·

Basic reporting

Authors have successfully responded to my comments.

Experimental design

None.

Validity of the findings

None.

Additional comments

None.

·

Basic reporting

The English language should be improved by a fluent speaker to ensure that an international audience can clearly understand your text. Some examples where the language could be improved include abstract (lines 36 to 37, 60 to 61), introduction part (lines 79 to 80), result part (lines 278 to 281) et al– the current phrasing makes comprehension difficult.
Table text should be labelled & described clearly. For example, the patient groups should be labelled as primary, metastasis et al in Table 1.

Experimental design

Ding et al investigated the gene mutation spectrum of ten patients with liver cancer in blood using 560 cancer-related gene panel by next generation of targeted sequencing technology. They compared gene mutation spectrum in patients with primary or secondary liver cancer, filtered pathogenic mutations, drew gene pathway network, and analyzed impacts of relevant genes on survival using data from public database. Their work is of clinical value and provides insights and novel contributions to the potential utility of ctDNA in diagnosis and management of liver cancer. Yet current study needs to be revised to be more concisely and straightforward throughout the result and discussion.

Validity of the findings

Improvement in next-generation sequencing technology and better understanding of genetic or epigenetic alteration of primary hepatocellular carcinoma have allowed comprehensive analysis of mutational landscape of ctDNA. Hotspot mutation panels have both shown promising performance which have been validated in current study as well. The survival analysis from GEPIA and HPA database have improved the significance of current study.

---

## Round 0.3 · accepted · Accept

Dear Dr. Zuo,

Thanks for your continued effort in revising your manuscript. The reviewers have recommended accepting it publication. Congratulations.

·

Basic reporting

The authors have addressed my questions nicely and provided sufficient data to support conclusions.

Experimental design

The primary research falls within the journal and research question is well defined. Yet there has been no effective treatment to HCC so far and current study could provide limited insights into clinical practice.

Validity of the findings

The authors provided sufficient data and the conclusion is well stated in the revised version.